# 1 Timely estimates of India's annual and monthly fossil $CO_2$ emissions

2 Robbie M. Andrew, CICERO Center for International Climate Research, Oslo, Norway,

3 robbie.andrew@cicero.oslo.no

## 4 Abstract

India is the world's third-largest emitter of carbon dioxide and is developing rapidly. While
India has pledged an emissions-intensity reduction as its contribution to the Paris
Agreement, the country does not regularly report emissions statistics, making tracking
progress difficult. Moreover, all estimates of India's emissions in global datasets represent
its financial year, not aligned to the calendar year used by almost all other countries. Here I
compile monthly energy and industrial activity data allowing the estimation of India's $CO_2$
emissions by month and calendar year with a short lag. Emissions show clear seasonal
patterns, and the series allows the investigation of short-lived but highly significant events,
such as the near-record monsoon in 2019 and the COVID-19 crisis in 2020. Data are available
at https://doi.org/10.5281/zenodo.3894394 (Andrew, 2020a).

Keywords: India, $CO_2$ emissions, Covid-19, seasonality

## 16 Introduction

As the world rapidly approaches the temperature limits set in the Paris Agreement
(CONSTRAIN, 2019), timely estimates of greenhouse gas emissions are critical for steering
policy and scientific understanding of the global carbon cycle (Le Quéré et al., 2020). India,
although having low per-capita emissions, is the world's third-largest emitter of carbon
dioxide (Friedlingstein et al., 2019), yet its most recent official report of emissions covers the
single year 2014 (GOI, 2018).

According to available estimates, India's $CO_2$ emissions have grown by about 5%/yr over
2010–2018 (Crippa et al., 2019). This growth has mainly been driven by expansion of the
economy as, among other things, the country's labour pool grows and much-needed energy
supply is increased (Karstensen et al., 2020), and much of this energy is supplied by coal and
petroleum products, including the transition from biomass to petroleum fuels, continuing
the long-term increase in the share of India's energy supplied from fossil fuels (see
Supplement Figure 42). Countering these upward pressures on $CO_2$ emissions, India's recent
development of variable renewables, particularly solar and wind, has exerted a downward
pressure on emissions growth, assisted by a sharp decline in prices for these technologies
and ambitious goals for renewables growth that have repeatedly been strengthened
(Khanna, 2010; MNRE, 2015; Varadhan, 2019). Development of variable renewables has
been further assisted by the introduction of reverse auctions and the creation of solar parks,
among other measures (Bose and Sarkar, 2019). In addition, the difficulty India has faced in
ramping up domestic coal production has probably also restrained emissions growth (Carl,
2015).

India does publish a great deal of energy data, but it is scattered across many documents,
often not in machine-readable form, occasionally containing errors, and generally without
much documentation. The country's official estimates of $CO_2$ emissions are infrequent and
never for more than a single year (GOI, 2004, 2012, 2015, 2018). Moreover, these reported

emissions are for India's financial year, running from April to March, so that they do not align with the calendar-year estimates provided by almost every other country (Andrew, 2020b). This gap in official reporting has been filled by third parties estimating emissions largely based on available financial-year publications, whether directly or via intermediate sources (e.g., IEA, 2019a; Gilfillan et al., 2019; EIA, 2020; Hoesly et al., 2018; GHG Platform India, no date), and not all of these are freely available.

Given the rapid pace both of India's development and of the change in global context, emissions estimates at a frequency greater than annual are also of interest. Higher-frequency data open up opportunities to analyse the relationships between emissions and policy shifts, economic cycles, weather, and more. The ability to explain why emissions have changed is critical to developing effective emissions policies.

Much of India's energy data is not available in formats that are readily machine-readable. In many cases, tables must be copied from PDF-format reports, either automatically using 'scraping' scripts, or by hand. On some occasions, reports posted on official websites are low-quality scans of signed documents, further reducing the availability of these data for analysis. Furthermore, explanations for data are often lacking in detail, and can conflict across different datasets for reasons that are not immediately apparent (see Supplement: Coal 'consumption').

The International Energy Agency in 2020 stated that the "Government of India should … Improve the collection, consistency, transparency and availability of energy data across the energy system at central and state government levels" (IEA, 2020b, p. 18). While government ministries responsible for publishing these data are making moves to improve the availability of more recent data, there are still obvious examples of copy-and-paste errors in spreadsheets, random misspellings, filename glitches, and even incorrect units given in the Energy Statistics yearbook. During the Covid-19 lockdown in India, the Central Electricity Authority stopped publishing daily generation reports for four weeks. Clearly much data work is still manual, and further automation will significantly improve India's ability to produce robust and timely estimates of fossil $CO_2$ emissions.

Monthly emissions estimates are also a core input to atmospheric inversion models (Oda et al., 2018). The standard approach taken in the literature to produce monthly emissions estimates is to use a temporal profile based on partial monthly activity data to temporally downsample annual emissions estimates. Three examples of this downsampling approach in the literature are the very first seasonal estimates made by Rotty (1987), CDIAC's gridded estimates (Andres et al., 2011) and EDGAR's temporal profiles (Crippa et al., 2020). Rotty (1987) and Andres et al. (2011), for example, used coal-fired power generation as a proxy for all coal consumption in India, while Crippa et al. (2020) used a proprietary database of activity data. EDGAR's monthly gridded dataset has no intra-annual variation for India (pers. comm., Matthew Jones, 10 July 2020).

Here I present a new dataset collating available information on India's monthly energy production and consumption, as well as cement production, and use this dataset directly to estimate India's monthly and calendar-year $CO_2$ emissions. In contrast to downsampling

techniques, the method used here provides accurate estimates of monthly $CO_2$ emissions in India.

## Materials and Methods

Fossil $CO_2$ emissions can be divided into four main source categories: coal, oil, natural gas, and carbonates (Friedlingstein et al., 2019). For the fossil fuels, estimates of monthly consumption are required, while for carbonates, production statistics are needed. In all cases, apparent energy consumption approximates true consumption, omitting some minor changes in stocks. For example, reported consumption of petroleum products is most likely supply to the market, with stocks at petrol stations and in vehicles not accounted for. A summary of the methodology is presented here, while full details are provided in the Supplement, including a table of individual data sources.

While monthly coal consumption by utility power stations is reported (CEA, various years-b), India does not report sub-annual total coal consumption, and apparent consumption must therefore be calculated using data on production, imports, exports, and stock changes. While these data are incomplete, they are sufficient to produce a reasonable estimate of monthly coal consumption. Importantly, the goal of this analysis is an estimate of $CO_2$ emissions from all oxidation of solid fossil fuels, rather than the more limited emissions from combustion for energy purposes, and this means it is unnecessary to separate out, for example, coking coal used in steel manufacture, which is oxidised rather than combusted.

The energy data sources used include revised, historical data from the Indian Bureau of Mines (2019), Ministry of Coal (various years-a), UN Statistics Division (2020); provisional and revised data from Coal India Limited (CIL, various years), Singareni Collieries Company Ltd (SCCL, various years), Ministry of Coal (various years-c, various years-b), and Ministry of Mines (Ministry of Mines, various years); power station stocks from Central Electricity Authority (CEA, various years-a, b); and international trade from the Directorate General of Commercial Intelligence and Statistics (DGCIS, 2020) and Department of Commerce (DOC, 2020), supplemented by recent provisional estimates reported by the media. While these data sources combined allow a good estimate of production and stock changes of hard coal with a lag of less than one month, lignite production data has a slightly longer lag, and simple extrapolation is used to complete the picture for the most recent month or two (Supplement Figure 15). It is assumed that the share of consumed coal that is not oxidised is negligible.

Monthly data on production and consumption of petroleum products are available from the Petroleum Planning and Analysis Cell (PPAC) of the Ministry of Petroleum and Natural Gas. In all four categories, revised data are always used when available in preference to provisional data. Since consumption data are available, the apparent consumption approach used for coal is not required for petroleum products. All products except for bitumen and lubricants are assumed to be fully oxidised; while it is known that some naphtha is used for production of durable commodities, this share is not known, but may be discoverable using data from the Annual Survey of Industries (MOSPI, no date).

PPAC also publishes monthly data on production, import, and supply of natural gas (PPAC, various years-a, b). Some data on consumption by sector are also published, and these are used to estimate the proportion of natural gas that is oxidised.

For carbonates, monthly data on clinker production are not available, so monthly cement production statistics are combined with a time-varying estimate of the clinker ratio to produce an estimate of monthly clinker production. Data on production of lime, glass, and ceramics were not available, and emissions from these carbonate sources are therefore omitted; India's second Biennial Update Report indicates these emissions combined contributed 1.9% of fossil $CO_2$ emissions in 2013–14 (GOI, 2018).

Once monthly energy consumption and clinker production estimates are available, these are converted to estimates of $CO_2$ emissions. For fossil fuels this requires first converting the consumption in physical units to energy units using information from the International Energy Agency (IEA) for coal and petroleum products (IEA, 2019c, b) and PPAC (no date) for natural gas, and then applying emission factors from the Intergovernmental Panel on Climate Change's 2006 guidelines (Gómez et al., 2006). For clinker production, the method of Andrew (2019) is followed to estimate emissions from physical production in tonnes.

For complete details of the methodology, data sources used, comparisons of provisional and revised energy data, comparisons of energy data from different sources, and more, see the Supplement. The monthly energy and cement data collated here are available at https://doi.org/10.5281/zenodo.3894394 (Andrew, 2020a).

## Results and Discussion

Following the described methods, I have assembled monthly $CO_2$ emissions estimates for coal, oil, natural gas, and cement for India (Figure 1). The available data and methodology allow estimation of emissions from coal from September 2008, oil from April 1998, natural gas from January 2009, and cement from April 2001.

Emissions from oxidation of coal form the largest share of the total, rising from about 61% in 2010 to 66% in 2014, before levelling off to about 65% in 2019. Peak monthly emissions to date were in March 2019 with 157 Mt $CO_2$ in the month. While emissions from coal grew at an average rate of 6.2%/yr over 2009–2018, in 2019 they stalled, as electricity demand dropped dramatically (Supplement Figure 41).

Emissions from oxidation of oil (petroleum products) are the next-largest source with about 25% of the total, reaching 50–60 Mt $CO_2$ per month in recent years. Emissions from natural gas and cement production are both about 5% of the total.

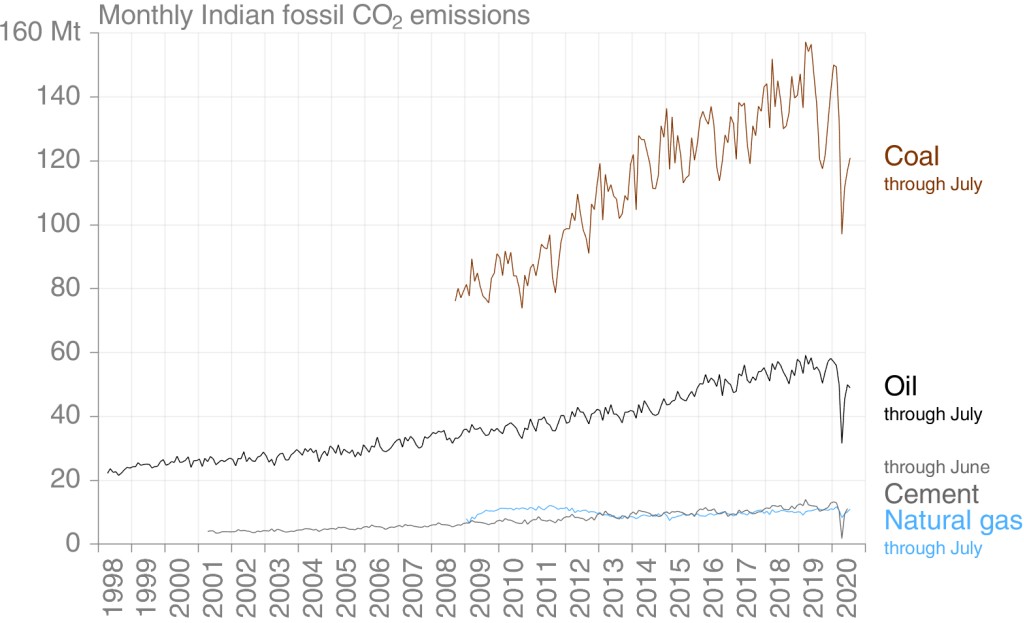

*Figure 1: Final monthly CO$_2$ emissions by category. Source: Own calculations.*

While the monthly emissions series appear quite volatile, X-11 seasonality analysis (Darné et
al., 2018; Shiskin et al., 1967; summarised in the Supplement) reveals strong, underlying
seasonal patterns (Figure 2). Coal emissions reach a peak in March through May, before
declining by up to 10% below the trend line for the typical southwest monsoon months of
June through August and then picking up again towards the end of the year, and emissions
from both oil and cement show similar though somewhat less smooth patterns. These
emissions patterns largely result from the effects of the monsoon's heavy rains, driving a
decline in industrial, construction and transportation activities. Coal emissions are also
driven down by the displacing effect of higher power generation from both hydropower and
wind during the monsoon season. In addition, oil emissions exhibit a consistent dip in
January and in March–April. Natural gas emissions show a substantially lower amplitude of
seasonality, under ±5%, with recent years showing a peak during the monsoon, apparently
driven by increased fertiliser production during these months. The seasonality of natural gas
emissions is also less stable over time, as supply constraints have changed considerably.
Despite relatively clear derived seasonal signals, considerable volatility is superimposed on
this seasonality in all emissions series (Figure 1).

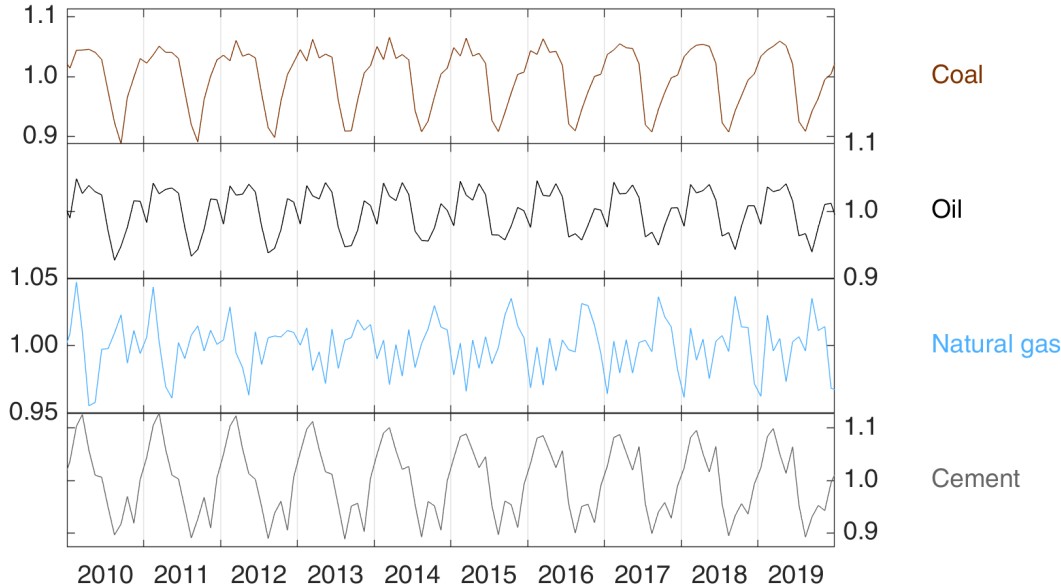

*Figure 2: Seasonality of the four emissions categories, derived using the X-11 method. Emissions in 2020 have been excluded*
*from the analysis because of their strong deviations from historical patterns.*

Turning to calendar-year emissions, Table 1 summarises emissions by category and total $CO_2$
emissions for India from 2009 to 2019. Each category has grown by 3%–5% per year over the
last five years, although growth from year to year has not been smooth, with coal emissions
stable in 2019. Total $CO_2$ emissions in India have grown from 1.6 Gt in 2009 to 2.6 Gt in 2019,
at an annual growth rate of 3.9%. (An equivalent table for financial-year emissions is
presented in the Supplement.) Over the same period, the $CO_2$-emissions intensity of India's
GDP has declined from 23.2 g/rupee to 18.1 g/rupee, about 2.2%/yr (Supplement Figure 40).

*Table 1: Calendar-year $CO_2$ emissions in India by category, million tonnes.*

| Year | Coal | Oil | Natural gas | Cement | Total |
|---|---|---|---|---|---|
| 2009 | 986 | 429 | 113 | 81 | 1608 |
| 2010 | 1019 | 435 | 134 | 86 | 1674 |
| 2011 | 1078 | 455 | 136 | 91 | 1762 |
| 2012 | 1226 | 485 | 126 | 100 | 1936 |
| 2013 | 1318 | 492 | 106 | 108 | 2023 |
| 2014 | 1447 | 507 | 107 | 116 | 2177 |
| 2015 | 1474 | 551 | 107 | 118 | 2249 |
| 2016 | 1541 | 609 | 113 | 123 | 2387 |
| 2017 | 1585 | 627 | 118 | 121 | 2451 |
| 2018 | 1670 | 651 | 123 | 139 | 2583 |
| 2019 | 1670 | 669 | 127 | 144 | 2609 |
| CAGR 2015-19* | 3.2% | 5.5% | 3.7% | 4.5% | 3.9% |

12  * Continuous compounding and adjusted for leap years.

13  The monthly Indian fossil $CO_2$ emissions dataset produced here includes all but about 2% of
14  anthropogenic fossil sources in the country, excluding emissions from decomposition of

fossil carbonates in the production of lime, glass and ceramics. The time lags of the emissions estimates are at most two months, and under one month for coal, the most important emissions source.

## Comparison with existing emissions estimates

To compare the emissions estimates produced here with other datasets, I aggregate monthly emissions to annual emissions against the Indian financial year, April–March. Figure 3 compares the emissions estimates produced here with those of the IEA (2019a) and CDIAC (Gilfillan et al., 2019), and also EDGAR (Crippa et al., 2019) for cement, noting that all three of these datasets report emissions in the period April 2017 through March 2018 as 2017 emissions.

For coal the method produces one series of oxidation emissions, and this is largely similar to the estimates from both IEA and CDIAC (Figure 3a). In the final two years 2017–18 and 2018–19, however, IEA has lower estimates. Close investigation has revealed potential errors in IEA's reported stock changes in both years, amounting to about 30 Mt in 2018-19 (detailed in Supplement section 2); IEA's 2018-19 estimate is indicated as being preliminary. Furthermore, IEA's data exclude changes of stocks at power stations, which exhibit large swings (Supplement Figure 10). CDIAC's estimate declines between 2012-13 and 2013-14, in strong contrast to the growths in other series, and this is because of CDIAC's use of UN energy data, which has a sharp drop in energy content of coal (see Supplement: Coal energy content).

For oil there are two series: combustion and oxidation (Figure 3b). The combustion series lies very close to that of the IEA – which specifically includes only energy uses of oil products – over the entire period. Oxidation emissions are on average about 50 Mt $CO_2$/yr higher throughout the period, largely reflecting emissions from oxidised naphtha and petroleum coke. CDIAC's series exhibits quite a different trend.

The natural gas emissions series includes three estimates: combustion, oxidation, and full oxidation (Figure 3c). The last of these assumes that all natural gas is oxidised, merely to present a bounding case. The combustion series agrees well with IEA's estimates, but again the CDIAC series exhibits a very different trend, diverging sharply from 2013–14. The oxidation series is significantly higher, largely reflecting the emissions from production and use of nitrogen-based fertilisers. Emissions show a very prominent peak in 2010–2012, a result of the rapid development of offshore gas field KG D6, but while this led to the construction of a number of gas-fired power stations, production from this field dropped substantially leading to greatly reduced domestic supplies and stranded power assets (MoP, 2019)(Supplement Figure 32).

For cement process emissions, the series is much lower than that of CDIAC, for reasons that have been explained elsewhere (Andrew, 2019). EDGAR's series appears to be reasonable up until 2010–11, when national clinker production data are readily available, but thereafter the trend appears unrealistic.

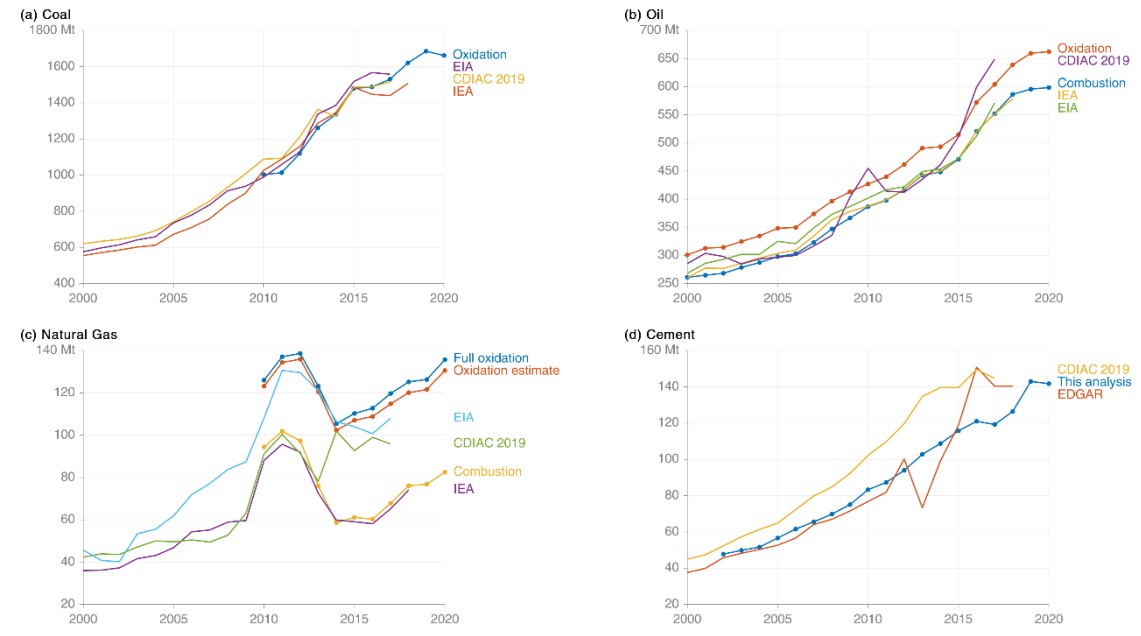

*Figure 3: Comparison of financial-year emissions estimates with other datasets, (a) coal, (b) oil, (c) natural gas, (d) cement.*
*Sources: (Gilfillan et al., 2019; IEA, 2019a; Crippa et al., 2019; EIA, 2020), own calculations.*

The most recent Indian official estimate of total $CO_2$ emissions was presented in India's
second biennial update report to the United Nations Framework Convention on Climate
Change (UNFCCC), with 1.998 Gt in the financial year 2013–14 (GOI, 2018). In the analysis
here, total $CO_2$ emissions in India in 2013–14 are estimated to be 2.04 Gt. While this is
strikingly close, this is not a true measure of the accuracy of the method since some errors
have cancelled: it is known that the emissions estimates generated here exclude some
carbonate sources, while emissions from naphtha oxidation here might be overestimated,
and there are other assumptions in various factors used here that introduce uncertainty.
Nevertheless, this match with the official total is encouraging.

## Deviations from forecasts

If official forecasts of growth in hard coal demand had played out, demand would have been
more than 20% higher in 2019–20, with consequently higher emissions (Figure 4). These
forecasts were based on assumptions of underlying growth in the economy of as much as
10%/yr (Ministry of Coal, 2011). In fact, the report on coal and lignite for the 12[th] five-year
plan included a second scenario with much higher demand growth, reaching 1200 Mt
already in 2016–17. While growth in demand followed the projection reasonably closely
until 2014–15, it has since slowed markedly.

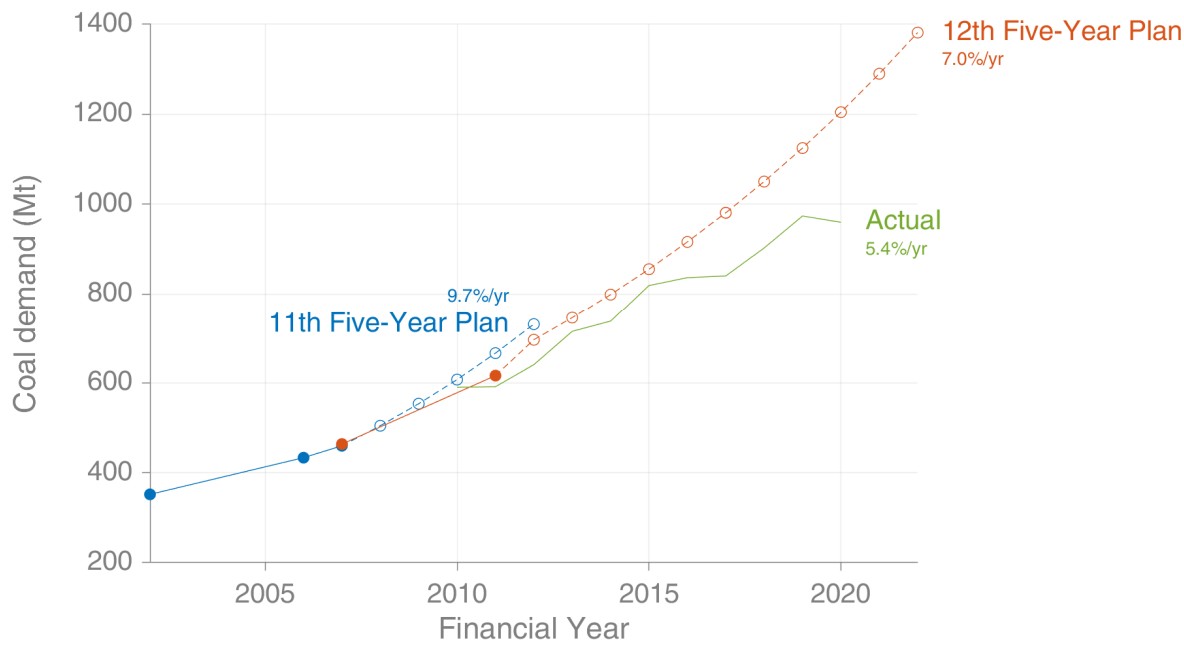

*Figure 4: India's hard coal demand. Filled circles and hollow circles show reported and projected demand in two five-year*
*plans, while the green line shows actual demand, and annual growth rates are indicated. Demand does not equal*
*consumption because of changes of stocks at power stations and industry. Source: Ministry of Coal (2006, 2011), own*
*calculations.*

There were certainly significant tailwinds in support of high growth in both demand and
supply of coal, such as strong political will, the 25%/yr annual average growth in coal imports
2011–2014, the rapid construction of new coal-fired capacity 2010–2016 (Supplement Figure
38), high targets for coal mining, the opening up of coal mining to competition, and
significant expansion of the labour pool, among others. But these faced an array of
headwinds constraining growth, including difficulty in acquiring land and environmental
permits, local protests, difficulty obtaining finance (CEA, 2019), rail under-capacity, debt,
subdued demand, unpredictable monsoon rains, "Coalgate" (illegal government coal block
allocations; Gilbert and Chatterjee, 2020), the dramatic fall in renewables prices, and large
economic shocks such as 2016's demonetisation, 2017's GST introduction, the shadow bank
crisis starting in 2018 (Subramanian and Felman, 2019), and 2020's COVID-19 pandemic. In
comparison, China's much larger consumption of coal grew by almost 9%/yr over 2000–2010
(NBS, 2019).

As suggested by Figure 5, growth in electricity generation from coal – recently about 75% of
all coal consumption – has been more linear than exponential in the last ten years.

Figure 5 also shows how significant the deviation in coal generation was in the latter half of
calendar-year 2019, also clear in Figure 1. From 2008 to 2018, the largest deviation of
monthly electricity generation with seasonality removed from the trend line is 110
24   GWh/day, while in 2019 it peaked at over 390 GWh/day. Generation from hydropower was
17% higher in 2019 than in 2018, partly a result of a very heavy southwest monsoon (IMD,
2019). But total electricity demand was down by almost 3% in the second half of 2019
compared to the same period in 2018, and more than 13% down in October 2019 (POSOCO,
2020), probably driven by a stalling economy (Subramanian and Felman, 2019), with value-

1 added growth in the manufacturing sector below zero in the period July to December 2019
2 (MOSPI, 2020).

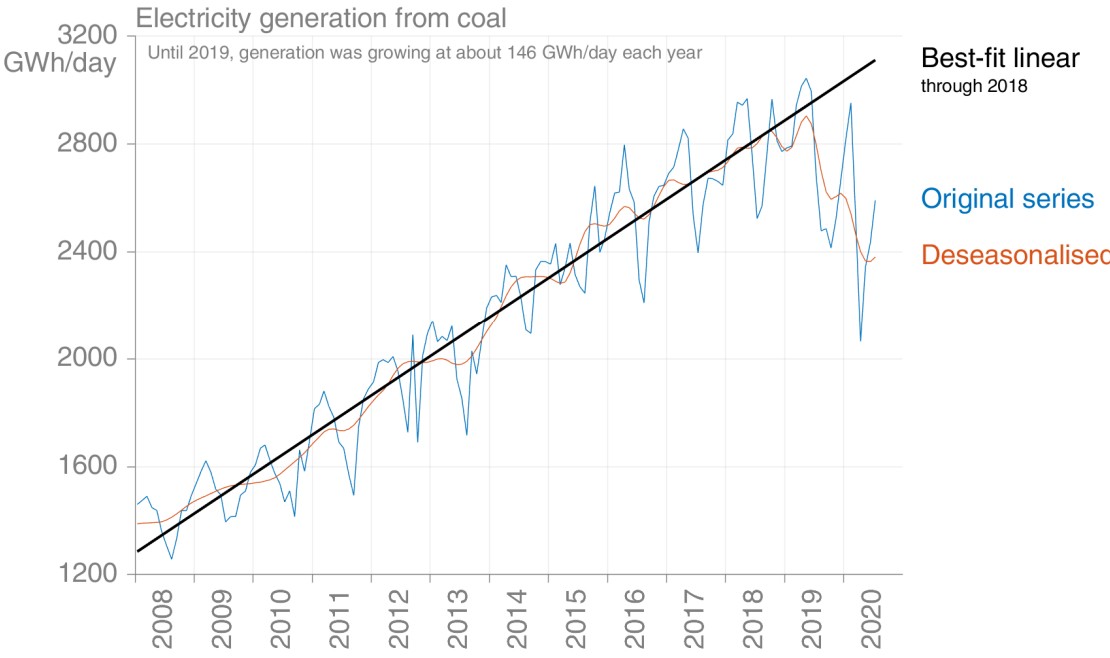

*Figure 5: Average daily electricity generation from coal by month, divided by month length, deseasonalised, and a best-fit*
*linear regression. Source: Central Electricity Authority, own calculations.*

The seasonal patterns of emissions generally follow the monsoon, particularly with less
electricity generation from coal. Rather than being due to decreased demand or difficulty
producing electricity from coal during the monsoon, the reason for this is higher generation
from both hydro and wind, both of which peak during the monsoon season. In fact,
electricity demand is highest through summer and lowest in winter (Supplement Figure 41),
with energy required in India for summer cooling substantially higher than energy required
for winter heating (Gaur et al., 2016).

COVID-19 effects
March would usually be one of the months of the year with the highest coal emissions
(Figure 2), but in 2020 this was affected by Covid-19 measures. India introduced a
nationwide lockdown (curfew) on March 25th, although some areas introduced lockdowns in
the days before (Roy and Phartiyal, 2020; Varadhan, 2020). Initially the lockdown was to be
for three weeks, but was repeatedly extended until the end of May (The Tribune, 2020), and
thereafter followed by a phased 'unlocking' (The Hindu, 2020). Largely as a result of
substantially reduced activity, and despite the lockdown only affecting about one-third of
the month, $CO_2$ emissions from coal in March 2020 were 15% lower than in March 2019
(Figure 6). But April saw the largest drops in emissions, with the lockdown having very
substantial effects on almost all areas of economic activity: total $CO_2$ emissions were down
40% compared to April 2019, with cement production dropping by 85%. Already in May
emissions started to rise again as constraints on activity were reduced, but the recovery is
far from complete, with July's consumption of oil products declining again compared to
June.

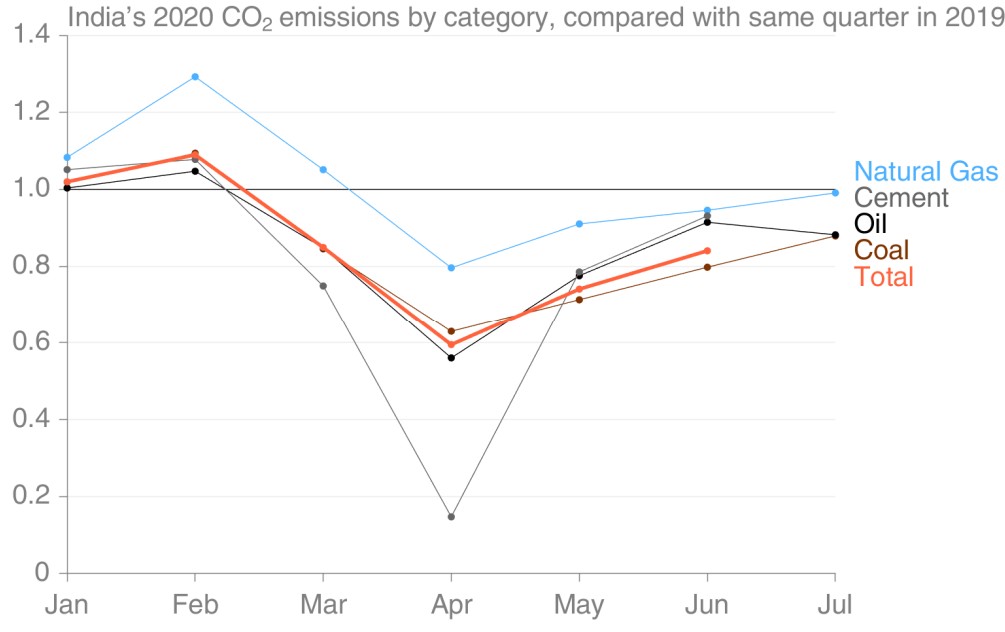

India's 2020 CO$_2$ emissions by category, compared with same quarter in 2019

Natural Gas
Cement
Oil
Coal
Total

*Figure 6: Quarter-on-quarter changes of CO$_2$ emissions by category during the first months of 2020.*

New approaches have recently been used to estimate the effect of the world's pandemic responses on global CO$_2$ emissions, based on collation of partial activity data for use as proxies, such as electricity generation data or travel indices (Le Quéré et al., 2020; IEA, 2020a; Liu et al., 2020). Such methods are suitable and useful for estimation of global effects in near real-time when more accurate and detailed data are not available, and the monthly estimates reported in the present work may be used to validate these alternative estimates of India's CO$_2$ emissions. Furthermore, given the close links between emissions of CO$_2$ and other air pollutants, studies on changes in air pollution due to India's lockdowns, could be cross-validated with the monthly CO$_2$ estimates reported here (e.g., Sharma et al., 2020; Mahato et al., 2020).

## Sources of uncertainty

There are several sources of uncertainty in these emissions estimates, which can be divided into four categories. First is the omission of some emissions sources. This analysis has excluded emissions from some carbonates, estimated to be equivalent to less than 2% of India's total CO$_2$ emissions. Further, some imported non-energy goods containing fossil carbon are excluded. While the case of Iceland shows clearly that imports of carbon anodes used in aluminium manufacture can be important (Andrew, 2020b), these are not imported by India (DGCIS, 2020). India does import urea from China, and the approach used here will not capture emissions from its use in agriculture; however, the amount is likely to be below 2 Mt/yr (see Supplement). A further missing source is that of low-temperature oxidation and spontaneous combustion of coal at mines, but available evidence suggests these would be significantly less than 1% of India's CO$_2$ emissions (Day et al., 2010; IPCC, 2019; Singh, 2019).

Second is use of provisional data and extrapolation before revised data are available. Revisions of coal, the most important emissions source, are in general relatively minor, and use of provisional data along with the methods used here to fill gaps are unlikely to

introduce significant error (see Supplement). Lignite production is relatively small and stable, so its extrapolation is not expected to introduce significant uncertainty. Moreover, if monthly press releases of mineral production have indeed recommenced, the lignite uncertainty will be largely removed in future, except for the most recent month(s).

Third is that of the revised data, effectively measurement error. While energy and emissions data in China serve as a cautionary example (Korsbakken et al., 2016), and India's economic production data face heavy revisions (Supplement Figure 39), these issues are not expected to affect India's energy and emissions data. One of the reasons for China's high data uncertainty is the very large number of enterprises involved, but in India energy and cement production are highly concentrated and closely monitored. As examples, two coal-mining companies, both state-owned, account for close to 90% of all coal production, and three state-owned fuel retailers account for about 90% of India's retail fuel sales (Reuters, 2020). While there have recently been claims of official tampering with economic statistics in India (Nadeem, 2019; The Telegraph, 2019), and incorrectly calculated productivity data (Singh, 2012), there is as yet no evidence of manipulation of energy or industrial production data.

The final category of uncertainty is in the emission factors, energy contents, and oxidised fractions used. This is perhaps the largest source of uncertainty, particularly the energy content of domestic hard coal, for which data have been scarce and inconsistent, and broad sampling efforts in recent years pointing to significant errors, with data from 2016-17 suggesting declared average coal quality was 10% higher than the true value (see Supplementary section: Coal energy content). More work is required to generate a more reliable time series of coal quality in India, but in the absence of additional historical sampling of coming to light, estimates will have to be made. A further source of uncertainty in this category is the assumption that all naphtha is oxidised, which potentially leads to an overestimate in the order of 1–2 $MtCO_2$/month.

The combination of data availability and assumptions made mean that coal emissions can be estimated with the shortest lag, within a week of the end of the month. Oil, natural gas, and cement emissions are usually delayed an additional month. There are two main reasons that coal emissions have a short lag. Firstly, coal-fired power stations have faced critical shortages at times and are monitored very closely, and secondly, the two largest mining companies, which report within a day of the month closing, make up the great majority of production. While short-lag emissions estimates require extrapolation of some components (e.g., lignite production), and use provisional data, as reported and revised data become available, these are incorporated into the estimation procedure used here.

While there are some identified deficiencies in the emissions estimates here, including the exclusion of emissions from use of limestone apart from in cement clinker production, comparisons with annual estimates from other sources, and in particular India's official reporting to the UNFCCC, suggests relatively good accuracy and therefore a high level of usefulness.

## Conclusions

India publishes more energy data than many other developing countries, providing a wealth of information for management, policy analysis and scientific research. Nevertheless, there

remains significant room for improvement in the quality of these publications. Possible avenues for such improvement include: (i) Publishing more data in machine-readable formats, rather than just as tables in PDF documents or in web-page tables, (ii) Providing a way for the public and researchers to ask questions about or report errors in data, establishing direct contact with those responsible for the data, to facilitate crowd-sourcing of quality assurance, (iii) Encouraging collaboration in data preparation and presentation across ministries to prevent errors creeping into reports, (iv) providing more documentation of reported data, (v) Reducing use of manual copy-pasting and typing, and automating as much as possible with both automatic and manual quality assurance, (vi) Standardising the use of important terms (e.g. 'consumption') across reports from different departments to prevent confusion, (vii) Making available older, non-electronic reports (e.g. Monthly Abstract of Statistics), online through use of digitisation.

The monthly, short-lag estimates of India's $CO_2$ emissions produced here will likely prove useful for tracking the country's progress against its nationally determined contribution under the Paris Agreement, but will also be useful for analysis of the drivers of India's emissions both historically and in future. Calendar-year estimates derived from these are also better aligned to the global datasets into which India's emissions are incorporated.

The future pathway of India's $CO_2$ emissions is highly uncertain. But India is developing rapidly in a world that – largely because of emissions in other countries – is carbon constrained. As India's population grows, as roads, railways and houses are built, as both vehicles and houses are electrified, as solar panels and wind turbines are installed, and as new coal mines are opened, tracking $CO_2$ emissions monthly will allow a closer observation on the consequences of these changes.

## Acknowledgements

This work was funded under the VERIFY project with funding from the European Union's Horizon 2020 research and innovation programme under Grant Agreement number 776810. The provision of data by the International Energy Agency for use in this work is gratefully acknowledged. Comments from three reviewers helped improve the manuscript.

## Data Availability
All monthly input data used in the analysis, in addition to the monthly emissions estimates, seasonality analysis results, and a copy of the Supplement, are available at https://doi.org/10.5281/zenodo.3894394 (Andrew, 2020a).

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
