# Peer review of "Timely estimates of India's annual and monthly fossil CO2 emissions"

_Earth System Science Data, 2020_

## Referee Comment (RC1) · Thomas Spencer (Referee) · 9 Jul 2020

Overall comments Overall, this is a useful and important paper, and of high scientific quality and policy importance.

It would be good to include a table with sources and hyperlinks if consistent with the journal policy.

From a policy-making point of view, I believe it would be useful for the paper to make a few observations on the ways in which India's data presentation could be best improved. Frustration with the disparate sources and poor presentation are widespread among the policy community and providing guidance on improvement would be valuable.

Page 1, Lines 24-27: It is worth mentioning here that India's carbon intensity of energy supply has also increased over the last 10-15 years, as the share of hydro electricity has declined, the share of coal increased, biomass transition in the residential sector has progressed, and emissions intensive fuels in industrial final energy consumption has increased. Page 1, Line 28: Better not to say "small renewables" as India has some of the largest utility scale solar parks in the world. Page 1, Line 40: this list of references should include the India GHG Platform initiative: http://www.ghgplatform-india.org/ Page 3, Line 25-27: it may be possible to get naptha consumption for production of durable commodities in the Annual Survey of Industries macro-data, and apply this ratio to monthly naptha consumption data. Page 3, Line 37-40: It is know that the calorific value of Indian coal varies greatly between different coal grades, and is generally understood to be declining over time as the quality of domestic mined coal declines. Some discussion of improved estimates of the calorific value of Indian coal should be made. Page 4, Lines 23-26, and Page 5, Line 1: it is worth noting that the observed monsoonal seasonality for coal, cement and oil is due in part to the same reason: economic activity in industry and construction declines during monsoon, implying reduced power demand and transport requirement. In additional residential electricity consumption declines as the temperature drops. Page 7: Lines 11-18: It would be good to discuss in a little detail, which errors may have cancelled. In addition, it would be good to explore the BUR to look at what emissions factors have been used for Indian coal, and how these compare with those used in this paper. Page 8: Lines 6 -16: This paragraph confused headwinds and tailwinds to coal supply with headwinds and tailwinds to coal demand. "difficulty in acquiring land and environmental permits, local protests, difficulty obtaining finance" relates to coal supply, while "large economic shocks such as 2016's demonetisation, 2017's GST introduction and 2020's COVID‐19 pandemic" relate to coal demand through channel of general macroeconomic growth. I believe the latter is much more important to understanding the deviation from forecast demand. In this regard, the paper could cite briefly some of the macroeconomic literature explaining India's growth

slowdown (for example: https://www.hks.harvard.edu/centers/cid/publications/faculty-working-papers/india-great-slowdown) Page 10, Lines 23-26: As discussed above, calorific value and emissions factor estimates for Indian coal may lead to significant uncertainties and are worth reviewing here.

---

## Referee Comment (RC2) · Anonymous Referee #2 · 15 Jul 2020

General comments: The paper addresses an extremely important aspect of the timeliness of India's GHG inventory reporting. Several data limitations and inconsistencies are rightly identified and an effort has been made to solve these, e.g. the differences in reporting intervals of coal production. The dataset provided here is extremely useful not just as activity data for CO2 emissions but also for other GHGs. Overall, the author has put in a great deal of effort into the paper and the supplement, which must be appreciated.

Specific comments: Attending to some of the concerns below, to the extent possible, might further enhance the usability of this dataset:

1. Page 1, Lines 27-31 note the role of renewable growth towards the stabilizing trend in CO2 emissions. It is also useful to point out here the opinion of some experts from

the literature that the shortage in coal production has taken due to a combination of complicated factors (land rights, political issues etc.). The following case study makes an excellent assessment of this, and I recommend 1-2 lines on such factors:

Carl, J. (2015). 4 The causes and implications of India's coal production shortfall. The Global Coal Market: Supplying the Major Fuel for Emerging Economies, 123-163.

2. In Page 1, Lines 38-40, it might be useful to point out (if applicable), that the third-party reporting through agencies by IEA might not be open-access and that adds to the utility of this dataset.

3. Page 3, Lines 3-9: I appreciate the explicitness in mentioning the difference in accounting only for combustion based emissions and overall oxidation. In the same vein, a line could be added here (or later) that future inventories could add additional emissions such as $CO_2$ emissions due to spontaneous emissions from coal mines; see following the reference and the recent 2019 IPCC Refinements:

Carras, J. N., Day, S. J., Saghafi, A., & Williams, D. J. (2009). Greenhouse gas emissions from low-temperature oxidation and spontaneous combustion at open-cut coal mines in Australia. International Journal of Coal Geology, 78(2), 161-168.

Singh, A. K. (2019). Better accounting of greenhouse gas emissions from Indian coal mining activities—A field perspective. Environmental Practice, 21(1), 36-40.

4. Page 3, Line 41 of main manuscript and section 6 of the Supplement: The authors note using the 2006 IPCC Guidelines default emission factors. However, Indian experts have developed national emission factors which have been vetted and included in the IPCC Emission Factor Database. I recommend using these emission factors either directly or atleast for a sensitivity analysis to look at the difference between default and country-specific emission factor.

5. With respect to Figures 12-14 of the supplement, is it possible to decompose the coal production further into surface- and underground-mined coal (either directly or

through % estimates from other sources)? That would make the dataset immediately usable for other applications such as methane estimation studies or life-cycle GHG studies.

6. In Page 5, line 4: Why does the author apportion the peaking of natural gas emission rise to use as peaking plants? I understand that the use of word "perhaps" conveys uncertainty but I welcome the author to convey the reason for their speculation.

7. Page 5, Lines 10-14 make important observations about variation of $CO_2$ emissions per year. The Government of India's INDC mentions its target as reduction of the GHG intensity (or GHG/GDP) by 33-35%. Therefore, in addition to comparing the GHG emissions, it might be useful to compare the $CO_2$ emissions per unit GDP as well to gauge consistency with the above goal.

8. Page 7, Lines 1-2 note the local peak due to KG-D6 basin. Additionally, do the authors have reason to believe that some emissions in the gas sector might have been due to the increase in coalbed methane production as well?

9. Page 7, Lines 3-4 mention stranded assets and it might also be useful to mention additional literature discussing potential stranded assets as climate restrictions come into force:

Malik, A., Bertram, C., Després, J., Emmerling, J., Fujimori, S., Garg, A., ... & Shekhar, S. (2020). Reducing stranded assets through early action in the Indian power sector. Environmental Research Letters, https://doi.org/10.1088/1748-9326/ab8033.

10. In Page 9, Lines 12-21 where authors point out the COVID-19 effects, it could also be mentioned that this dataset could be used as a correlation to the top-down effects on air pollution reported for Indian studies. This, in my view, further enforces the need for such a dataset.

11. Page 10, Lines 2-4 make an interesting point about imported urea use. Is it fair to assume that this is another reason why the emissions data in the paper track with the

government data as it is also the case for the UNFCCC data reporting practices?

---

## Referee Comment (RC3) · Charles Worringham (Referee) · 24 Jul 2020

This paper is a timely consideration of a significant issue: the status, quality, timeliness and implications of India's greenhouse emissions data, with a very useful summary of the compoinent contributions and recent trends. It makes a valuable contribution and underscores the importance of making data available in a regular manner with sufficient detail to ensure its accuracy and reliability.

Specific comments:

Page 1 line 28: Given the size of some of the utility-scale PV plants, the term "small renewables" should be re-considered.

Page 1 lines 29-30: It might be worth including India's reverse auctions and innovations

such as round-the-clock tenders in the list of contributory factors in renewables growth.

Page 1 lines 32-33: In addition to the limitations of the available data listed, it could be noted that official documents are maintained by different ministries and departments, which can also lead to outright inconsistencies between different official sources, such as disagreement between the CEA and the MOSPI Energy Statistics publication concerning the quantity of coal consumed for electricity generation in the three most recent years.

Page 3 line 3: The first statement should be qualified to acknowledge that monthly coal consumption figures for power generation are provided by the CEA.

Page 4 line 22 to page 5 line 7: The discussion of the seasonality of emissions is very important as it is such a strong factor in India's data. Accordingly, some extension of the discussion might be worthwhile, for example, to consider how monsoonal weather affects a) production and supply, hindered by weather affecting logistics, b) demand and consumption, for example decreased construction activity or abrupt decreases in air conditioning load as rains relieve extreme heat conditions that normally occur in May. In addition, the substantial increase in hydroelectric and wind generation that accompanies the monsoon and suppresses coal consumption could be noted here as well as on page 9. The benefit of an extended discussion is that it would guide readers who may wish to analyse seasonal changes in emissions in terms of other economic and meteorological data.

Page 6 lines 16-18: The apparent omission of power plant coal stockpile changes in the Energy Statistics publication might also be mentioned.

Page 10 lines 12-22: This paragraph considers data revisions and errors. Although it is correctly stated that coal statistics from CIL and SCCL undergo only minor corrections, it might be noted that data from captive power plants and other users are much more sporadic and provided only in summary form. Capturing total coal consumption could benefit from more systematic and timely data on non-CIL and non-SCCL data.

Page 11, lines 1-12. The Conclusions are entirely appropriate and relevant. Given the importance of timely and accurate data, and the multiple shortcomings noted in the body of the paper, a useful addition to this section could be a brief set of key recommendations that could guide efforts to better coordinate, accelerate and improve India's collation and publication of energy and related statistics.

───────────────────────

---

## Author Response (AR1)

**Referee #1**

Overall comments Overall, this is a useful and important paper, and of high scientific quality and policy importance.

It would be good to include a table with sources and hyperlinks if consistent with the journal policy.

Yes, good idea.

- Changes made: I have added a table to the Supplement, "India activity data sources."

From a policy-making point of view, I believe it would be useful for the paper to make a few observations on the ways in which India's data presentation could be best improved. Frustration with the disparate sources and poor presentation are widespread among the policy community and providing guidance on improvement would be valuable.

- Changes made: Added paragraph to the Conclusions: "India publishes more energy data than many other developing countries, providing a wealth of information for management, policy analysis and scientific research. Nevertheless, there remains significant room for improvement in the quality of these publications. Possible avenues for such improvement include: (i) Publishing more data in machine-readable formats, rather than just as tables in PDF documents or in web-page tables, (ii) Providing a way for the public and researchers to ask questions about or report errors in data, establishing direct contact with those responsible for the data, to facilitate crowd-sourcing of quality assurance, (iii) Encouraging collaboration in data preparation and presentation across ministries to prevent errors creeping into reports, (iv) providing more documentation of reported data, (v) Reducing use of manual copy-pasting and typing, and automating as much as possible with both automatic and manual quality assurance, (vi) Standardising the use of important terms (e.g. 'consumption') across reports from different departments to prevent confusion, (vii) Making available older, non-electronic reports (e.g. Monthly Abstract of Statistics), online through use of digitisation."

Page 1, Lines 24-27: It is worth mentioning here that India's carbon intensity of energy supply has also increased over the last 10-15 years, as the share of hydro electricity has declined, the share of coal increased, biomass transition in the residential sector has progressed, and emissions intensive fuels in industrial final energy consumption has increased.

- Changes made: I have added a clause to the end of this sentence.
  "…including the transition from biomass to petroleum fuels, continuing the long-term increase in the share of India's energy supplied from fossil fuels (see SI Fig 41)"

[Figure]

where the new figure in the Supplement is as below

Page 1, Line 28: Better not to say "small renewables" as India has some of the largest utility scale solar parks in the world.

- Changes made: Changed "small renewables" to "variable renewables".

Page 1, Line 40: this list of references should include the India GHG Platform initiative: http://www.ghgplatformindia.org/

- Changes made: added a reference to the GHG Platform India.

Page 3, Line 25-27: it may be possible to get naptha consumption for production of durable commodities in the Annual Survey of Industries macro-data, and apply this ratio to monthly naptha consumption data.

I don't have access to this, so I've simply suggested it as a possibility.

- Changes made: Added text ", but may be discoverable using data from the Annual Survey of Industries (MOSPI, no date)"

Page 3, Line 37-40: It is known that the calorific value of Indian coal varies greatly between different coal grades, and is generally understood to be declining over time as the quality of domestic mined coal declines. Some discussion of improved estimates of the calorific value of Indian coal should be made.

I have tried to keep most of the detail of the methods in the Supplement, since they're so extensive. I have added the following text and figure to the Supplement.

- Changes made: Added to Supplement:

  Focusing on hard coal, Figure 17 compares a number of different datasets, demonstrating wide divergence in reported coal quality. It seems clear that coal quality overall has declined in the last 50 years, partly as a result of the significant increase in the share of lower-cost production from open-cast mines (77% in 1998/99 to 94% in 2018/19, according to the Coal

Directories), but the IEA's figures in the 1970s and 1980s are markedly different from those reported in all but the most recent Energy Statistics yearbooks.

It is unclear how the Energy Statistics derives average coal quality, but it appears that the IEA has used the annual data on production by coal grade, combined with average energy contents for each grade. This supposition is based on the author doing exactly that with the data provided by the Coal Directories: from 2013, estimates made this way match very closely to those of the IEA. Before 2013, India used a less-detailed grading system. The author's estimates for that earlier period assume that the average energy content did not jump dramatically upwards from 2012 to 2013, something that seems unlikely, and this leads to a difference with the IEA's estimates in that period.

In 2016, Coal India introduced quality assurance routines, sending samples to third-party laboratories for assessment of energy content, a scheme called 'Unlocking Transparency by Third Party Assessment of Mined Coal' (UTTAM). This scheme was introduced after repeated complaints by power station operators that received coal was of lower than the declared (and paid-for) energy content. With 51% sampling coverage in the 2017-18 year, UTTAM results showed that the average analysed energy content was 6% lower than the average declared energy content. Back-calculation of energy content from hard coal production in both energy and mass terms suggests that the Energy Statistics report has subsequently simply used this much lower average for the entire period reported (2006-07 through 2018-19 in the 2020 edition).

The UN Statistics Division's Energy Yearbooks report much higher energy contents in 2012 and 2013, with these numbers having been reported to them by Indian officials; subsequent values are taken from IEA reports (pers. comm., Leonardo Rocha Souza, 16 July 2020). This sharp drop in the UN data for India's energy content translates directly into a sharp drop in production from 2012-13 to 2013-14, which propagates directly to CDIAC's estimates of emissions from solid fuels for India.

Given the insufficient sampling until the introduction of the UTTAM scheme in 2016, it is impossible to say with any uncertainty what the energy content of India's hard coal was before then, but it is unlikely that the constant low value used by the Energy Statistics yearbook is correct.

[Figure]

*Figure 17: Comparison of energy content of Indian hard coal from various datasets. Data plotted for the Coal Directory ('CoalDir') are the author's estimates derived from data on production by grade. IEA WEB/WES is the World Energy Balances (energy units) and World Energy Statistics (mass units).*

Page 4, Lines 23-26, and Page 5, Line 1: it is worth noting that the observed monsoonal seasonality for coal, cement and oil is due in part to the same reason: economic activity in industry and construction declines during monsoon, implying reduced power demand and transport requirement. In additional residential electricity consumption declines as the temperature drops.

Yes, good point, although I don't see a reduction in total electricity demand during the monsoon season. In 2015–2018 it wasn't until October or November that demand dropped as winter temperatures arrived. A drop in residential consumption because of reduced AC use is presumably offset by increases in other sectors' demand. See the figure towards the end of the Supplement.

- Changes made: Added the following sentences: "These emissions patterns largely result from the effects of the monsoon's heavy rains, driving a decline in industrial, construction and transportation activities. Coal emissions are also driven down by the displacing effect of higher power generation from both hydropower and wind during the monsoon season."

Page 7: Lines 11-18: It would be good to discuss in a little detail, which errors may have cancelled. In addition, it would be good to explore the BUR to look at what emissions factors have been used for Indian coal, and how these compare with those used in this paper.

With regard to error cancellation, I already had the following text "it is known that the emissions estimates generated here exclude some carbonate sources" and add a further clause for some additional information.

- Changes made: Added clause "while emissions from naphtha oxidation here might be overestimated"

As for the BUR, I have added the following paragraph to the section in the Supplement that discusses energy contents.

- Changes made: Added paragraph to Supplement:

  Emissions from coal in India's Second Biennial Update Report (BUR) are derived using country-specific energy contents and emission factors (GOI, 2018). The Report is unclear as to whether these factors, reported in tables 2.3 and 2.4, are only used for domestic coal, or whether they are averages for total coal supply, including imports. Imported coal is of higher quality than India's domestic coal, and this likely explains why the energy contents provided in table 2.3 for coking and non-coking coal (23.66 and 18.26 MJ/kg, respectively) are somewhat higher than those reported by the IEA for domestic coal (20.50 and 16.69 MJ/kg). The BUR's reported energy content of lignite, which is entirely domestic, is 9.80 MJ/kg, very similar to the IEA's 9.55 MJ/kg, and somewhat lower than the Energy Statistics' value of 11.37 MJ/kg.

Page 8: Lines 6 -16: This paragraph confused headwinds and tailwinds to coal supply with headwinds and tailwinds to coal demand. "difficulty in acquiring land and environmental permits, local protests, difficulty obtaining finance" relates to coal supply, while "large economic shocks such as 2016's demonetisation, 2017's GST introduction and 2020's COVID-19 pandemic" relate to coal demand through channel of general macroeconomic growth. I believe the latter is much more important to understanding the deviation from forecast demand. In this regard, the paper could cite briefly some of the macroeconomic literature explaining India's growth slowdown (for example: https://www.hks.harvard.edu/centers/cid/publications/facultyworking-papers/india-great-slowdown)

I do agree that this paragraph discusses headwinds and tailwinds of both supply and demand, but that was in fact intentional. However, since the previous paragraph is very specifically about demand, this transition was not made clear to the reader.

- Changes made:
  Added "in both demand and supply of coal" to the first sentence of the paragraph.
  Added "the shadow bank crisis starting in 2018 (Subramanian and Felman, 2019)"

Page 10, Lines 23-26: As discussed above, calorific value and emissions factor estimates for Indian coal may lead to significant uncertainties and are worth reviewing here.

- Changes made: Added text: "This is perhaps the largest source of uncertainty, particularly the energy content of domestic hard coal, for which data have been scarce and inconsistent, and broad sampling efforts in recent years pointing to significant errors, with data from 2016-17 suggesting declared average coal quality was 10% higher than the true value (see Supplementary section: Coal energy content). More work is required to generate a more reliable time series of coal quality in India, but in the absence of additional historical sampling of coming to light, estimates will have to be made."

**Referee #2**

General comments: The paper addresses an extremely important aspect of the timeliness of India's GHG inventory reporting. Several data limitations and inconsistencies are rightly identified and an effort has been made to solve these, e.g. the differences in reporting intervals of coal production. The dataset provided here is extremely useful not just as activity data for CO2 emissions but also for other GHGs. Overall, the author has put in a great deal of effort into the paper and the supplement, which must be appreciated.

Specific comments: Attending to some of the concerns below, to the extent possible, might further enhance the usability of this dataset:

1. Page 1, Lines 27-31 note the role of renewable growth towards the stabilizing trend in CO2 emissions. It is also useful to point out here the opinion of some experts from the literature that the shortage in coal production has taken due to a combination of complicated factors (land rights, political issues etc.). The following case study makes an excellent assessment of this, and I recommend 1-2 lines on such factors:

Carl, J. (2015). 4 The causes and implications of India's coal production shortfall. The Global Coal Market: Supplying the Major Fuel for Emerging Economies, 123-163.

I believe the point the reviewer suggests I make is largely already made later, in the section "Deviation from forecasts", where I say "But these faced an array of headwinds constraining growth, including difficulty in acquiring land and environmental permits, local protests, difficulty obtaining finance (CEA, 2019), rail under-capacity, debt, subdued demand, unpredictable monsoon rains, "Coalgate" (illegal government coal block allocations; Gilbert and Chatterjee, 2020), the dramatic fall in renewables prices, and large economic shocks such as 2016's demonetisation, 2017's GST introduction, the shadow bank crisis starting in 2018, and 2020's COVID-19 pandemic."

Nevertheless, I agree that the point is usefully made already in the introduction.

- Changes made: Added sentence "In addition, the difficulty India has faced in ramping up domestic coal production has probably also restrained emissions growth (Carl, 2015)."

2. In Page 1, Lines 38-40, it might be useful to point out (if applicable), that the thirdparty reporting through agencies by IEA might not be open-access and that adds to the utility of this dataset.

- Changes made: Added text ", and not all of these are freely available"

3. Page 3, Lines 3-9: I appreciate the explicitness in mentioning the difference in accounting only for combustion based emissions and overall oxidation. In the same vein, a line could be added here (or later) that future inventories could add additional emissions such as CO2 emissions due to spontaneous emissions from coal mines; see following the reference and the recent 2019 IPCC Refinements:

Carras, J. N., Day, S. J., Saghafi, A., & Williams, D. J. (2009). Greenhouse gas emissions from low-temperature oxidation and spontaneous combustion at open-cut coal mines in Australia. International Journal of Coal Geology, 78(2), 161-168.

Singh, A. K. (2019). Better accounting of greenhouse gas emissions from Indian coal mining activities: A field perspective. Environmental Practice, 21(1), 36-40.

This is very interesting. Certainly, future official inventories should attempt to include estimates of emissions from both low-temperature oxidation and any resulting spontaneous combustion of coal. The Australian paper the reviewer cites concludes that $CO_2$ emissions from spontaneous combustion in the open-cast mines sampled ranged between 0.01% and 1.34% of the amount of $CO_2$ from eventual combustion of the mines' produced coal. These are very small amounts.

The IPCC 2019 Refinements provide several default factors. Using the average factor for surface mining, 0.44 $m^3CO_2$/tonne of coal produced, would result in emissions that are 0.03% of India's total $CO_2$ emissions. The paper the reviewer cites by Singh reports a higher value, from sampling three Indian mines, but this would still result in emissions less than 0.5% of India's total.

Given the very small magnitude of these emissions sources, it's unlikely that I would include these in future revisions of this dataset, when there are much larger sources of uncertainty in this analysis. But I will add this to the section on uncertainty in the estimates.

- Changes made: Added a sentence in the discussion of sources of uncertainty: "A further missing source is that of low-temperature oxidation and spontaneous combustion of coal at mines, but available evidence suggests these would be significantly less than 1% of India's $CO_2$ emissions (Day et al., 2010; IPCC, 2019; Singh, 2019)"

4. Page 3, Line 41 of main manuscript and section 6 of the Supplement: The authors note using the 2006 IPCC Guidelines default emission factors. However, Indian experts have developed national emission factors which have been vetted and included in the IPCC Emission Factor Database. I recommend using these emission factors either directly or at least for a sensitivity analysis to look at the difference between default and country-specific emission factor.

Indeed, India's reporting to the UNFCCC is based on national coal energy contents and emission factors. However, the information provided in the BUR is insufficient to make use of these, and there has been considerable change over time, not reported in the BUR, which is only for a single year. For now, I will retain the factors I have used, but these can readily be changed in future revisions of this dataset. I have added some discussion of these factors, in addition to significantly extending the discussion of coal energy content in the Supplement.

- Changes made: Added paragraph to Supplement:

  Emissions from coal in India's Second Biennial Update Report (BUR) are derived using country-specific energy contents and emission factors (GOI, 2018). The Report is unclear as to whether these factors, reported in tables 2.3 and 2.4, are only used for domestic coal, or whether they are averages for total coal supply, including imports. Imported coal is of higher quality than India's domestic coal, and this likely explains why the energy contents provided in table 2.3 for coking and non-coking coal (23.66 and 18.26 MJ/kg, respectively) are somewhat higher than those reported by the IEA for domestic coal (20.50 and 16.69 MJ/kg). The BUR's reported energy content of lignite, which is entirely domestic, is 9.80 MJ/kg, very similar to the IEA's 9.55 MJ/kg, and somewhat lower than the Energy Statistics' value of 11.37 MJ/kg.

5. With respect to Figures 12-14 of the supplement, is it possible to decompose the coal production further into surface- and underground-mined coal (either directly or through % estimates from other sources)? That would make the dataset immediately usable for other applications such as methane estimation studies or life-cycle GHG studies.

Yes, that would be useful. Unfortunately, I've found no data that show this split on a sub-annual basis. The *Coal Directory* has data on the share of open-cast and underground production, available from fiscal year 1999, but I have found no monthly breakdown of these. One could make some assumptions, but I have no information on, for example, whether underground mines are less or more affected by the monsoon than open-cast mines. I note that the IEA in its 2020 energy data edition has introduced a fugitive dataset, including methane emissions from coal mining, reaching 1435 kt in 2018.

- Changes made: none.

6. In Page 5, line 4: Why does the author apportion the peaking of natural gas emission rise to use as peaking plants? I understand that the use of word "perhaps" conveys uncertainty but I welcome the author to convey the reason for their speculation.

I'm glad you questioned this. Spurred by this comment, I have looked more closely. The monthly data available since 2014 show no indication of any consistent seasonal cycle in natural gas usage for power consumption, so this suggestion of peaking plants was mistaken. However, fertiliser production, which accounts for about a third of natural gas consumption, does exhibit a distinct seasonal cycle, peaking in monsoon months, presumably in response to agricultural demand. I have reworded the text as follows:

- Changes made: Replaced "perhaps being used as peaking plants" with "apparently driven by increased fertilizer production during these months"

7. Page 5, Lines 10-14 make important observations about variation of CO2 emissions per year. The Government of India's INDC mentions its target as reduction of the GHG intensity (or GHG/GDP) by 33-35%. Therefore, in addition to comparing the GHG emissions, it might be useful to compare the CO2 emissions per unit GDP as well to gauge consistency with the above goal.

- Changes made:
  Added the following figure to the Supplement:

[Figure]

  Added a sentence to the manuscript: "Over the same period, the $CO_2$-emissions intensity of India's GDP has declined from 23.2 g/rupee to 18.1 g/rupee, about 2.2%/yr (Supplement Figure 40)."

8. Page 7, Lines 1-2 note the local peak due to KG-D6 basin. Additionally, do the authors have reason to believe that some emissions in the gas sector might have been due to the increase in coalbed methane production as well?

Data from the Ministry of Petroleum and Natural Gas's Indian Petroleum & Natural Gas Statistics 2018-19 provide CBM production since 2008, and while production has increased from 13 MMSCM to a high so far of 735 MMSCM in 2017/18, the peak proportion of CBM in total natural gas production was only 2.3%, smallest (<0.5%) during the years of growth in offshore production. So no, CBM is not significant here.

- Changes made: none.

9. Page 7, Lines 3-4 mention stranded assets and it might also be useful to mention additional literature discussing potential stranded assets as climate restrictions come into force:

Malik, A., Bertram, C., Després, J., Emmerling, J., Fujimori, S., Garg, A., ... & Shekhar, S. (2020). Reducing stranded assets through early action in the Indian power sector. Environmental Research Letters, https://doi.org/10.1088/1748-9326/ab8033.

This looks like an interesting article, thank you. When I raise the point about stranded assets, it is as a sort of footnote in the context of the substantial rise and fall of natural gas production. The issue of stranded power station assets in general is much larger, and not something I think I should go into in this paper. Unfortunately, Malik et al don't mention the stranding of natural gas fired power stations, otherwise it would have been logical to add this reference.

- Changes made: none.

10. In Page 9, Lines 12-21 where authors point out the COVID-19 effects, it could also be mentioned that this dataset could be used as a correlation to the top-down effects on air pollution reported for Indian studies. This, in my view, further enforces the need for such a dataset.

Yes, I agree.

- Changes made: Added sentence "Furthermore, given the close links between emissions of $CO_2$ and other air pollutants, studies on changes in air pollution due to India's lockdowns, could be cross-validated with the monthly $CO_2$ estimates reported here (e.g., Sharma et al., 2020; Mahato et al., 2020)."

11. Page 10, Lines 2-4 make an interesting point about imported urea use. Is it fair to assume that this is another reason why the emissions data in the paper track with the government data as it is also the case for the UNFCCC data reporting practices?

Yes, in this paper I'm trying to include all significant sources of $CO_2$ emissions in India. In India's reporting to the UNFCCC they are required to do this, and the IPCC Guidelines for inventory construction include such details as calculating emissions from use of urea. The IPCC approach to emissions from use of urea in agriculture is agnostic to where the urea is made, and instead calculates emissions from all urea purchased by agriculture.

- Changes made: none.

**Referee #3**

This paper is a timely consideration of a significant issue: the status, quality, timeliness and implications of India's greenhouse emissions data, with a very useful summary of the compoinent contributions and recent trends. It makes a valuable contribution and underscores the importance of making data available in a regular manner with sufficient detail to ensure its accuracy and reliability.

Specific comments:

Page 1 line 28: Given the size of some of the utility-scale PV plants, the term "small renewables" should be re-considered.

- Changes made: Changed "small renewables" to "variable renewables".

Page 1 lines 29-30: It might be worth including India's reverse auctions and innovations such as round-the-clock tenders in the list of contributory factors in renewables growth.

I'll avoid reference to round-the-clock tenders, since there's some indication that the implementation of these does not reflect the name: https://economictimes.indiatimes.com/industry/energy/power/round-the-clock-renewable-energy-tenders-worry-developers/articleshow/76012308.cms

- Changes made: Added sentence "Development of variable renewables has been further assisted by the introduction of reverse auctions and the creation of solar parks, among other measures (Bose and Sarkar, 2019)."

Page 1 lines 32-33: In addition to the limitations of the available data listed, it could be noted that official documents are maintained by different ministries and departments, which can also lead to outright inconsistencies between different official sources, such as disagreement between the CEA and the MOSPI Energy Statistics publication concerning the quantity of coal consumed for electricity generation in the three most recent years.

I've taken some time to look at this specific inconsistency. MoSPI's Energy Statistics for coal consumption by the electricity sector (table 6.4 in the 2020 edition) are taken from the Ministry of Coal, and, since 2010, are identical to the numbers in the Coal Directories (table 4.20 in the 2018-19 edition), except for the final year, which comes from the Provisional Coal Statistics. These data represent *despatches of domestic coal to both utility and captive power generators*, not consumption at all, despite the title of both the chapter and table. It seems imported coal used by power stations is included in the 'Others plus import non-coking' column, partly explaining why this column has such large values. The supply data they use do not allow disaggregation of non-coking coal imports by using sector. Nor does this table account for stock changes at power stations. Meanwhile, the CEA data only include utility generation, not captive. So to reconcile the data in these tables one must take the utility despatch data from the Coal Directory (or PCS) and the total coal receipts less imports from CEA. These two are approximately the same, with some residual as is common with comparison of supply and use data from different sources.

The annual Energy Statistics from MoSPI is severely lacking in descriptive text, making it very difficult to determine what the data really mean.

- Changes made: Added a section to the Supplement describing this specific point on coal consumption, and referred to this section in the Introduction as "Furthermore, explanations for data are often lacking in detail, and can conflict across different datasets for reasons that are not immediately apparent (see Supplement: Coal 'consumption')"

Page 3 line 3: The first statement should be qualified to acknowledge that monthly coal consumption figures for power generation are provided by the CEA.

- Changes made: Reworded to "While monthly coal consumption by utility power stations is reported (CEA, various years-b), India does not report sub-annual total coal consumption, and apparent consumption must therefore be calculated using data…"

Page 4 line 22 to page 5 line 7: The discussion of the seasonality of emissions is very important as it is such a strong factor in India's data. Accordingly, some extension of the discussion might be worthwhile, for example, to consider how monsoonal weather affects a) production and supply, hindered by weather affecting logistics, b) demand and consumption, for example decreased construction activity or abrupt decreases in air conditioning load as rains relieve extreme heat conditions that normally occur in May. In addition, the substantial increase in hydroelectric and wind generation that accompanies the monsoon and suppresses coal consumption could be noted here as well as on page 9. The benefit of an extended discussion is that it would guide readers who may wish to analyse seasonal changes in emissions in terms of other economic and meteorological data.

I agree. I've added some of these ideas to the discussion here.

- Changes made: Added the following sentences: "These emissions patterns largely result from the effects of the monsoon's heavy rains, driving a decline in industrial, construction and transportation activities. Coal emissions are also driven down by the displacing effect of higher power generation from both hydropower and wind during the monsoon season."

Page 6 lines 16-18: The apparent omission of power plant coal stockpile changes in the Energy Statistics publication might also be mentioned.

- Changes made: Added sentence: "Furthermore, IEA's data exclude changes of stocks at power stations, which exhibit large swings (SI Fig 10)."

Page 10 lines 12-22: This paragraph considers data revisions and errors. Although it is correctly stated that coal statistics from CIL and SCCL undergo only minor corrections, it might be noted that data from captive power plants and other users are much more sporadic and provided only in summary form. Capturing total coal consumption could benefit from more systematic and timely data on non-CIL and non-SCCL data.

While I would readily have agreed with this statement at the time it was made, I have since discovered that the Ministry of Coal has recently started reporting provisional year-to-date production and offtake explicitly including both captive and other mines at https://coal.nic.in/content/production-and-supplies. I have added a module to my code to check this page and calculate the differences to give monthly production, and will include these in my published dataset. The Internet Archive provided the website in May 2020, allowing the estimation of June's production by difference. I hope the publication of these year-to-date data will continue, but there's never any guarantee.

Moreover, there is also the Monthly Summary to Cabinet (https://coal.nic.in/content/monthly-summary-cabinet), which has included a statement on the production of captive mines since the September 2017 edition. I have added these to my published dataset.

The more general point about having to fill in gaps in available data is made in the preceding paragraph.

- Changes made: none.

Page 11, lines 1-12. The Conclusions are entirely appropriate and relevant. Given the importance of timely and accurate data, and the multiple shortcomings noted in the body of the paper, a useful addition to this section could be a brief set of key recommendations that could guide efforts to better coordinate, accelerate and improve India's collation and publication of energy and related statistics.

[revised manuscript text omitted]

The landscape for Indian activity data is composed of historical data sources, stable ongoing data sources, and unstable sources for low-lag data. National, revised monthly coal production data are reported by the Indian Bureau of Mines with a lag of more than six months, and at time of writing over 12 months (Indian Bureau of Mines, 2019). Provisional national coal and lignite production data were published with a lag of less than two months via press release by the Ministry of Mines until mid-2017 (Ministry of Mines, 2017), but these were not released for about 18 months, reappearing in March 2020 with provisional data for January 2020, although these data are of low precision, and their publication remains unreliable (Ministry of Mines, 2020). The Ministry of Coal has recently begun publishing total provisional fiscal-year-to-date national hard coal production, broken down by CIL, SCCL, Captive, and Other (Ministry of Coal, no date), and with regular access, these can be converted to monthly production values. The Coal Controller's Organisation (CCO) at the Ministry of Coal produces an annual report called Provisional Coal Statistics (PCS) that include monthly national coal production, with a lag of about 7-9 months (Ministry of Coal, various years-c). The CCO also publishes revised statistics in the Coal Directory, with a lag of about 12-16 months (Ministry of Coal, various years-b). The United Nations Statistics Division's 'Monthly Bulletin of Statistics Online' also includes monthly coal production for India (UN Statistics Division, 2020). Lastly, the now-discontinued Monthly Abstract of Statistics was published by the Central Statistics Organisation (now Ministry of Statistics and Programme Implementation) (CSO/MoSPI, various years). This last dataset appears to be available for earlier years going back several decades, but the author has not been able to obtain access to these editions. These  datasets are compared and their availability by month shown in Figure 1 (the datasets are so similar that mostly they lie atop one another in the figure). While all figures here are for hard coal, all five of the data sources also report lignite production.

While national coal production data are lacking in recent months, the two largest coal mining companies, Coal India Limited (CIL) and Singareni Collieries Company Limited (SCCL), release their provisional monthly production and offtake data in the first days of the following month (CIL, various years; SCCL, various years). These two companies represent about 90% of Indian coal production. Reporting of data on provisional production at captive mines has recently been introduced in the Ministry of Coal's Monthly Summary to Cabinet (Ministry of Coal, various years-a) and also on the Ministry's website as year-to-date data, which also reports the provisional small production from other mines (Ministry of Coal, no date). In the two months for which all data are available (Sep 2017 and Jan 2020), the sum of provisional production data from CIL, SCCL and captive mines is within 2% of the provisional national production figure, demonstrating that this sum is suitable to fill the gap in provisional national production when production from other mines is not available.

Revised coal production data are available from CIL both in their provisional production reports, which compare to the same (revised) month in the previous year, and in their more recent quarterly reports. In the available data, CIL's revisions are generally within 0.25% of provisional statistics, except for one anomalous data point in 2016 that was revised by 0.7% (Figure 2). For SCCL, available data show that revisions are also within 0.25% of provisional data (Figure 3). No revised data for captive production are available. When the sum of provisional data from CIL, SCCL and captive mines are compared with revised national production, the latter is always higher in the period where data are available, although always less than 2.5% higher (Figure 4), representing the production of a small number of other mines

.

[Figure]

*Figure 1: Comparison and availability of the  datasets of India's monthly national coal production. Sources: Monthly Abstracts of Statistics, Coal Directories, Provisional Coal Statistics , 
[revised manuscript text omitted]

[Figure]

Based on IEA data from the IEA (2018) World Energy Statistics,
www.iea.org/statistics. All rights reserved.

*Figure 13: India's supply of coal. Source: IEA, MoSPI Yearbooks, monthly data assembled herein.*

For some countries, imports of coal-derived non-energy products such as carbon anodes used in aluminium smelting are significant (Andrew, 2020), but no data was found to suggest this in India.

[Figure]

*Figure 14: Monthly imports of coal by type. Source: Department of Commerce*

**4. Extrapolation**

Lignite production data lag behind data on production of hard coal and must be extrapolated.

[Figure]

*Figure 15: Extrapolation of lignite production. Line with circle markers shows reported values, while line without markers shows interpolation/extrapolation.*

**5. Coal energy content**

The Indian Government introduced quality sampling of coal from 2016 (ETEnergyWorld, 2016), but while these data are collected throughout the year, they are only available on a cumulative basis. India's Energy Yearbook provides tables of annual production and imports in both physical and energy units, but these deviate significantly from those used by the IEA (Figure 16). Here I choose to use the energy contents from the IEA (2019c, 2019d), assuming its information is more reliable, particularly for earlier years.

[Figure]

Based on IEA data from the IEA (2018) World Energy Balances, www.iea.org/statistics. All rights reserved.

*Figure 16: Comparison of energy content of coal from IEA (2019c, 2019d) and India's Energy Yearbooks (MOSPI, various years).*

Focusing on hard coal, Figure 17 compares a number of different datasets, demonstrating wide divergence in reported coal quality. It seems clear that coal quality overall has declined in the last 50 years, partly as a result of the significant increase in the share of lower-cost production from open-cast mines (77% in 1998/99 to 94% in 2018/19, according to the Coal Directories), but the IEA's figures in the 1970s and 1980s are markedly different from those reported in all but the most recent Energy Statistics yearbooks.

It is unclear how the Energy Statistics derives average coal quality, but it appears that the IEA has used the annual data on production by coal grade, combined with average energy contents for each grade. This supposition is based on the author doing exactly that with the data provided by the Coal Directories: from 2013, estimates made this way match very closely to those of the IEA. Before 2013, India used a less-detailed grading system. The author's estimates for that earlier period assume that the average energy content did not jump dramatically upwards from 2012 to 2013, something that seems unlikely, and this leads to a difference with the IEA's estimates in that period.

In 2016, Coal India introduced quality assurance routines, sending samples to third-party laboratories for assessment of energy content, a scheme called 'Unlocking Transparency by Third Party Assessment of Mined Coal' (UTTAM). This scheme was introduced after repeated complaints by power station operators that received coal was of lower than the declared (and paid-for) energy content. With 51% sampling coverage in the 2017-18 year, UTTAM results showed that the average analysed energy content was 6% lower than the average declared energy content. Back-calculation of energy content from hard coal production in both energy and mass terms suggests that the Energy Statistics report has subsequently simply used this much lower average for the entire period reported (2006-07 through 2018-19 in the 2020 edition).

The UN Statistics Division's Energy Yearbooks report much higher energy contents in 2012 and 2013, with these numbers having been reported to them by Indian officials; subsequent values are taken from IEA reports (pers. comm., Leonardo Rocha Souza, 16 July 2020). This sharp drop in the UN data for India's energy content translates directly into a sharp drop in production from 2012-13 to 2013-14, which propagates directly to CDIAC's estimates of emissions from solid fuels for India.

Given the insufficient sampling until the introduction of the UTTAM scheme in 2016, it is impossible to say with any uncertainty what the energy content of India's hard coal was before then, but it is unlikely that the constant low value used by the Energy Statistics yearbook is correct.

[Figure]

*Figure 17: Comparison of energy content of Indian hard coal from various datasets. Data plotted for the Coal Directory ('CoalDir') are the author's estimates derived from data on production by grade. IEA WEB/WES is the World Energy Balances (energy units) and World Energy Statistics (mass units).*

Emissions from coal in India's Second Biennial Update Report (BUR) are derived using country-specific energy contents and emission factors (GOI, 2018). The Report is unclear as to whether these factors, reported in tables 2.3 and 2.4, are only used for domestic coal, or whether they are averages for total coal supply, including imports. Imported coal is of higher quality than India's domestic coal, and this likely explains why the energy contents provided in table 2.3 for coking and non-coking coal (23.66 and 18.26 MJ/kg, respectively) are somewhat higher than those reported by the IEA for domestic coal (20.50 and 16.69 MJ/kg). The BUR's reported energy content of lignite, which is entirely domestic, is 9.80 MJ/kg, very similar to the IEA's 9.55 MJ/kg, and somewhat lower than the Energy Statistics' value of 11.37 MJ/kg.

**6. Coal CO$_2$ emissions**

I calculate apparent hard coal and lignite consumption in energy terms separately as production + net imports + net withdrawal from stocks. These are then converted to CO$_2$ emissions using default factors from the IPCC's guidelines (Gómez et al., 2006). Resulting monthly emissions estimates are shown in Figure 18.

[Figure]

Figure 18: Final monthly estimates of CO$_2$ emissions from oxidation of coal in India.

**7. Coal 'consumption'**

Two official Indian reports provide data on coal consumption by the electricity sector. But the numbers they report disagree significantly. The problem is the absence of any definition of 'consumption' in the *Energy Statistics*.

MoSPI's *Energy Statistics* publication presents coal consumption by the electricity sector (table 6.4 in the 2020 edition), with a footnote indicating the source is "Office of the Coal Controller, Ministry of Coal). Since 2010, these data are identical to the numbers in the Coal Controller's *Coal Directory* reports (table 4.20 in the 2018-19 edition), except for the final year, which comes from the *Provisional Coal Statistics*. Importantly, these data represent despatches of domestic coal to both utility and captive power generators, not consumption at all, despite the title of both the chapter and table in *Energy Statistics*. It seems imported coal used by power stations is included in the 'Others plus import non-coking' column, partly explaining why this column has such large values. The supply data they use from the Coal Controller do not allow disaggregation of non-coking coal imports by using sector. Nor does this table account for stock changes at power stations. Meanwhile, the Central Electricity Authority's monthly *Coal Statements* only include consumption by utility generation, not captive generation. Therefore, to reconcile the data in these tables one must take the utility despatch data from the *Coal Directory* (or *PCS*) and the total coal receipts less imports from the *Coal Statements*. These two are approximately the same, with some residual as is common with comparison of supply and use data from different sources.

**7.8. Petroleum production and consumption**

Consumption data by mass are available for 12 different petroleum products including non-energy uses such as bitumen, starting in April 1998 (Figure 19)(PPAC, various years-a). These data are most likely in fact sales data rather than actual consumption, a distinction that gains more significance when looking at monthly as opposed to annual data.

[Figure]

*Figure 19: Consumption of petroleum products from April 1998 in physical units. Source: PPAC.*

To convert to units of energy I  use factors from the IEA (2018b), which are similar but not identical to the IPCC default factors (Gómez et al., 2006).

Since this analysis focusses on India's domestic emissions, fuel consumption by international aviation and navigation (i.e. bunkers) are excluded. The consumption data from PPAC exclude marine bunker fuels but include aviation bunker fuels, the same convention used by the IEA in its Oil Demand tables (IEA, 2019a). I use the annual ratio of bunker to non-bunker consumption from IEA (2018a) to estimate and remove monthly aviation bunker fuels. This effectively assumes, for example, that the proportion of jet kerosene supplying international flights is constant through the year.

The resulting consumption data in energy units are shown in Figure 20.

[Figure]

*Figure 20: Consumption of petroleum production from April 1998 in energy units. Source: Own calculations.*

To determine combustion emissions, non-energy uses of petroleum products must be removed. IEA data also indicate non-energy use by fuel type; these vary gradually over time, and I assume the fractions in the final year of the IEA data also apply for the years immediately following. For oxidation, I assume that both bitumen and lubricants are never oxidised, but that all other fuels are. This is likely to be a small overestimate because some naphtha and other petroleum products are used as feedstocks to produce commodities that might never oxidise. The resulting energy dataset is converted to $CO_2$ emissions using default IPCC factors (Gómez et al., 2006).

[Figure]

*Figure 21: Emissions from combustion of petroleum products, excluding refinery emissions. Source: Own calculations.*

Lastly, emissions from refineries' own use of energy are added by scaling annual refinery energy use in the form of petroleum products from IEA (2018a) to monthly production data available from April 2010 (PPAC, various years-b). The IEA indicate that energy use from petroleum products by refineries is entirely refinery gas (IEA, 2019c), and emissions are therefore determined using the default IPCC emission factor for refinery gas (Gómez et al., 2006). Where monthly production data are not available, annual production data are used to estimate refinery emissions. This assumption introduces a small month-to-month error, but refinery emissions are small compared to total petroleum emissions.

[Figure]

Based on IEA data from the IEA (2018) World Energy Balances, www.iea.org/statistics. All rights reserved.

*Figure 22: Emissions from combusted petroleum products: Source: Own calculations.*

[Figure]

Based on IEA data from the IEA (2018) World Energy Balances, www.iea.org/statistics. All rights reserved.

*Figure 23: Emissions from oxidised petroleum products: Source: Own calculations.*

[Figure]

*Figure 24: Final monthly estimates of CO₂ emissions from oxidation of oil and oil products in India.*

The Joint Organisations Data Initiative (JODI) publishes monthly data on oil and gas production and consumption for a large number of countries, but when comparing India's total oil demand with the official, revised data series from PPAC, some considerable deviations are evident (Figure 25).

[Figure]

*Figure 25: Comparison of monthly oil demand from PPAC and JODI.*

**Comparison with IEA annual consumption data**

The following figures demonstrate that the monthly consumption data as used match very closely the annual data provided by the IEA.

[Figure]

Based on IEA data from the IEA (2018) World Energy Statistics,
www.iea.org/statistics. All rights reserved.

*Figure 26: Comparison of consumption of diesel, gasoline, LPG, naphtha, and jet kerosene in physical units between aggregated monthly data from PPAC and annual data from IEA.*

[Figure]

Based on IEA data from the IEA (2018) World Energy Balances,
www.iea.org/statistics. All rights reserved.

*Figure 27: Comparison of consumption of diesel, gasoline, LPG, naphtha, and jet kerosene in energy units between aggregated monthly data from PPAC and annual data from IEA.*

[Figure]

Based on IEA data from the IEA (2018) World Energy Statistics, www.iea.org/statistics. All rights reserved.

*Figure 28: Comparison of consumption of fuel oil, lubricants, and other kerosene in physical units between aggregated monthly data from PPAC and annual data from IEA.*

[Figure]

Based on IEA data from the IEA (2018) World Energy Balances, www.iea.org/statistics. All rights reserved.

*Figure 29: Comparison of consumption of fuel oil, lubricants, and other kerosene in energy units between aggregated monthly data from PPAC and annual data from IEA.*

[Figure]

Based on IEA data from the IEA (2018) World Energy Balances,
www.iea.org/statistics. All rights reserved.

*Figure 30: Comparison of consumption of petroleum coke, bitumen, and other oil products in physical units between aggregated monthly data from PPAC and annual data from IEA.*

[Figure]

Based on IEA data from the IEA (2018) World Energy Statistics,
www.iea.org/statistics. All rights reserved.

[revised manuscript text omitted]

IEA: World Energy Balances 2020 Edition, International Energy Agency, Paris, 2020. www.iea.org (Last access: 17 August 2020).

Indian Bureau of Mines: Monthly Statistics of Mineral Production, 2019. http://ibm.nic.in/index.php?c=pages&m=index&id=497 (Last access: 26 November 2019).

Ministry of Coal: Production and Supplies, no date. https://coal.nic.in/content/production-and-supplies (Last access: 11 August 2020).

Ministry of Coal: Monthly Summary for Cabinet, various years-a. https://coal.nic.in/content/monthly-summary-cabinet (Last access: 26 November 2019).

Ministry of Coal: Coal Directory of India, Coal Controller's Organisation, Ministry of Coal, Kolkata, various years-b. http://www.coalcontroller.gov.in/pages/display/16-coal-directory (Last access: 17 April 2020).

Ministry of Coal: Provisional Coal Statistics, Coal Controller's Organization, Ministry of Coal, Kolkata, various years-c. http://www.coalcontroller.gov.in/pages/display/20-provisional-coal-statistics (Last access: 17 April 2020).

Ministry of Mines: Mineral Production during August 2017 (Provisional), 2017. https://pib.gov.in/newsite/PrintRelease.aspx?relid=172036 (Last access: 28 October 2019).

Ministry of Mines: Mineral Production during January 2020 (Provisional), 2020. https://pib.gov.in/newsite/PrintRelease.aspx?relid=200538 (Last access: 31 March 2020).

MOSPI: Energy Statistics, Central Statistics Office, Ministry of Statistics and Programme Implementation, various years. http://www.mospi.gov.in/recent-reports (Last access: 26 August 2019).

OEA: Eight Core Industries, Office Of The Economic Adviser, Department For Promotion Of Industry And Internal Trade, 2019. https://eaindustry.nic.in/ (Last access: 26 August 2019).

POSOCO: National Load Despatch Centre: Daily Reports, Power System Operation Corporation Limited, 2020. https://posoco.in/reports/daily-reports/ (Last access: 16 April 2020).

PPAC: Conversion Factors, Petroleum Planning & Analysis Cell, Ministry of Petroleum and Natural Gas, no date. https://www.ppac.gov.in/content/232_2_Others.aspx (Last access: 17 April 2020).

PPAC: Consumption of Petroleum Products, Petroleum Planning & Analysis Cell, Ministry of Petroleum & Natural Gas, various years-a. https://www.ppac.gov.in/content/146_1_ProductionPetroleum.aspx (Last access: 20 April 2020).

PPAC: Production of Petroleum Products by Refineries & Fractionators, Petroleum Planning & Analysis Cell, Ministry of Petroleum & Natural Gas, various years-b. https://www.ppac.gov.in/content/146_1_ProductionPetroleum.aspx (Last access: 20 April 2020).

SCCL: Provisional Production and Dispatches Performance of SCCL, The Singareni Collieries Company Limited, various years. https://scclmines.com/scclnew/performance_production.asp (Last access: 17 April 2020).

UN Statistics Division: Monthly Bulletin of Statistics Online, United Nations Statistics Division, 2020. https://unstats.un.org/unsd/mbs/app/DataSearchTable.aspx (Last access: 17 April 2020).